# A simple Turing reaction–diffusion model explains how PLK4 breaks symmetry during centriole duplication and assembly

**Zachary M. Wilmott**[1,2], **Alain Goriely**[2]*, **Jordan W. Raff**[1]*

**1** Sir William Dunn School of Pathology, University of Oxford, Oxford, United Kingdom, **2** Mathematical Institute, University of Oxford, Oxford, United Kingdom

\* goriely@maths.ox.ac.uk (AG); jordan.raff@path.ox.ac.uk (JWR)

**Data Availability Statement:** All relevant data are within the paper and its Supporting Information files.

## Abstract

Centrioles duplicate when a mother centriole gives birth to a daughter that grows from its side. Polo-like-kinase 4 (PLK4), the master regulator of centriole duplication, is recruited symmetrically around the mother centriole, but it then concentrates at a single focus that defines the daughter centriole assembly site. How PLK4 breaks symmetry is unclear. Here, we propose that phosphorylated and unphosphorylated species of PLK4 form the 2 components of a classical Turing reaction–diffusion system. These 2 components bind to/unbind from the surface of the mother centriole at different rates, allowing a slow-diffusing activator species of PLK4 to accumulate at a single site on the mother, while a fast-diffusing inhibitor species of PLK4 suppresses activator accumulation around the rest of the centriole. This "short-range activation/long-range inhibition," inherent to Turing systems, can drive PLK4 symmetry breaking on a either a continuous or compartmentalised Plk4-binding surface, with PLK4 overexpression producing multiple PLK4 foci and PLK4 kinase inhibition leading to a lack of symmetry-breaking and PLK4 accumulation—as observed experimentally.

## Introduction

Most human cells are born with a single pair of centrioles comprising an older "mother" and a younger "daughter" (Fig 1A); these organelles play an important part in many aspects of cellular organisation [1–4]. The centriole pair duplicates precisely once during each cell division cycle when the original centriole pair separate, and a single new daughter grows off the side of each preexisting centriole (now both termed mothers) [5–7]. The centriole is a 9-fold symmetric structure and it is unclear how its symmetry is broken to establish the single site for daughter centriole assembly. Polo-like-kinase 4 (PLK4) is the master regulator of centriole biogenesis [8,9], and it appears to be initially recruited around the entire surface of the mother centriole before it becomes concentrated at a single focus that defines the daughter centriole assembly site (Fig 1B) [6,10–12].

Two different mathematical models have previously been proposed to explain how PLK4 symmetry might be broken. The approach developed by Kitagawa and colleagues [13], was

**Funding:** The research was funded by a Wellcome Trust Senior Investigator Award (215523) to J.W.R (and supporting ZW). https://wellcome.org/ The funders had no role in study design, data collection and analysis, decision to publish, or preparation of the manuscript.

**Competing interests:** The authors have declared that no competing interests exist.

motivated by the observation that PLK4 has an intrinsic ability to self-organise into macromolecular condensates in vitro [14–16], and that PLK4 and its centriole receptor CEP152 initially appear to be organised into discrete compartments around the centriole—approximately 12 for CEP152 and approximately 6 for PLK4 [13]. The authors assume that PLK4 initially binds equally well to the 12 CEP152 compartments and that, due to its self-assembling properties, this PLK4 then recruits additional PLK4. They allow PLK4 to autoactivate itself as the local concentration of PLK4 rises, and suppose that active PLK4 is more mobile in the condensates than inactive PLK4 (as observed experimentally). Thus, the active PLK4 generated in one compartment can activate PLK4 in nearby compartments, which stimulates the PLK4 to leave the nearby compartment. In this way, each of the approximately 12 CEP152 compartments effectively compete to recruit PLK4, which can then "laterally inhibit" recruitment at nearby compartments. This process allows the approximately 12 CEP152 compartments to generate a "pre-pattern" of approximately 6 PLK4 compartments. At the start of S-phase, the addition of

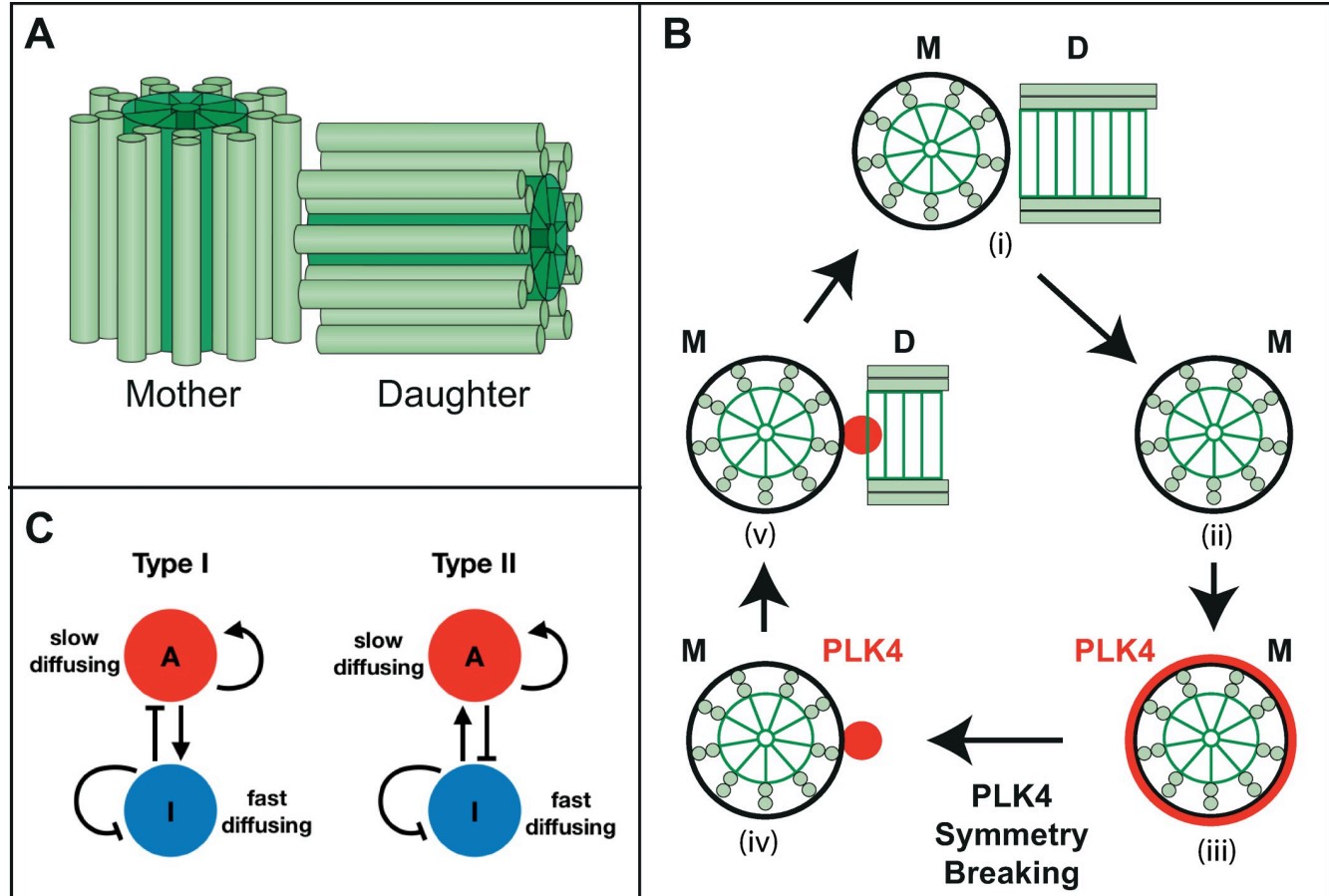

**Fig 1. Schematic illustration of centrioles, centriole duplication, PLK4 symmetry breaking, and Turing reaction–diffusion systems. (A)** Most higher-eukaryotic cells are born with a single pair of centrioles comprising an older mother and a younger daughter formed around a 9-fold symmetric central cartwheel (dark green) surrounded by 9-sets of microtubule doublets or triplets (light green), depending on the species. The centrioles depicted here are from *Drosophila*, which have doublet MTs and which lack the distal and sub-distal appendages found on human mother centrioles. **(B)** The centriole pair (i) duplicates once each cell cycle when the mother (M) and daughter (D) separate (ii) (for simplicity only the original mother is shown here); the daughter matures into a mother (not shown) and both mother centrioles recruit PLK4 (*red*) symmetrically around themselves (iii). PLK4 symmetry is broken (iv) when PLK4 becomes concentrated in a single spot that defines the site of new daughter centriole assembly (v). PLK4 usually dissociates from the centriole by the time daughter centriole assembly is complete [74]. **(C)** Diagrams illustrate the logic of the 2 chemical reaction schemes that can form Turing reaction–diffusion systems (see text for details).

STIL into the system generates additional positive feedback and ensures that the pre-patterned site with most PLK4 becomes the single dominant site.

In an alternative approach developed by Goryachev and colleagues [17], the centriolar surface is also modelled as a number of discrete compartments but, unlike the Takao and colleagues model, these are not spatially ordered (so all compartments are equivalent and any chemical species produced at one compartment can diffuse equally to all the other compartments). These compartments bind cytoplasmic PLK4, but also the key centriole-assembly protein STIL. PLK4 and STIL interact both in the cytoplasm and in the centriole compartments, with exchange between the centriole compartments occurring via the shared cytoplasm. The phosphorylation of STIL by PLK4 is postulated to increase the centriolar retention of the phosphorylated PLK4:STIL complexes. Since PLK4 can promote its own activation [18–20], this system forms a positive feedback loop in which PLK4 activity locally auto-amplifies itself by phosphorylating PLK4:STIL complexes to strengthen their centriolar retention. This process creates a competition between the different compartments for binding to the various PLK4: STIL complexes. This system can break symmetry to establish a single dominant compartment that concentrates the phosphorylated PLK4:STIL complexes.

The different mathematical frameworks used in the 2 models make it difficult to compare them, so it is unclear if there are any similarities in the logic of the underlying biochemical reactions that determine the behaviour of each system and that can explain how 2 such apparently different models can both break symmetry. Moreover, although both models can account for certain aspects of PLK4 symmetry breaking, neither provide a complete description. For example, the 12-fold symmetry of the CEP152 Receptor and 6-fold symmetry of the PLK4 pre-pattern described by Takao and colleagues have not been observed in other systems [21–24]. This model also explicitly relies on STIL appearing in the system only after PLK4 has already broken symmetry; this may be plausible in somatic cells, but not in rapidly dividing systems such as the early *Drosophila* embryo where the cytoplasmic concentration of Ana2/STIL remains constant through multiple rounds of centriole duplication [25]. The Leda and colleagues model predicts that inhibiting PLK4 kinase activity should effectively deplete PLK4 from the centriole surface, but it is now clear that kinase-inhibited PLK4 accumulates at centrioles to much higher levels than normal [15,26,27].

Here, we model PLK4 symmetry breaking as a "Turing system," which we define as a two-component reaction–diffusion system that breaks symmetry through activator–inhibitor dynamics [28,29]. Turing systems are famous for their ability to produce complex spatial patterns, and they have been well-studied in relation to numerous biological phenomena including the formation of animal coat patterns, predator–prey dynamics, and the spread of disease [30–33]. These systems rely on "short-range activation/long-range inhibition," whereby relatively fast chemical reactions between the 2 components initially drive the system towards a steady state, which can then be destabilised by the differential diffusion of the 2 components to drive pattern-forming instabilities over a longer timescale.

Although the 2 previous models of PLK4 symmetry breaking appear to be very different, we show that both can be reformulated as Turing systems, with phosphorylated/non-phosphorylated species of PLK4 (either on its own, or in a complex with STIL) forming the two components in the system that bind/unbind from centrioles at different rates (allowing the two components to effectively differentially diffuse within the system). Through computer simulations we demonstrate that these simple models can break PLK4 symmetry to form a single PLK4 peak, while overexpressing PLK4 can lead to the formation of multiple PLK4 peaks, and inhibiting PLK4 kinase activity can lead to the uniform accumulation of PLK4 around the centriole surface—as observed experimentally. Unexpectedly, our analysis reveals that, in the appropriate parameter regime, the dominant chemical reactions that drive symmetry breaking

in the two previous models are actually the same. Potentially importantly, and in contrast to the previous models, our models can support PLK4 symmetry breaking on either a continuous PLK4-binding surface or on a surface comprising any number of discrete Plk4- binding compartments.

## Results

We initially model the recruitment of PLK4 around the surface of the centriole as a two-component reaction–diffusion system acting on a one-dimensional ring (the centriole surface). It may be shown mathematically that the probabilistic random walk of a molecule that repeatedly binds to, and then unbinds from, a surface on the microscale acts as a diffusive process on the macroscale [34]. Therefore, we use surface diffusion as a simplifying modelling approximation for diffusion through the bulk, where the effective diffusion rate of a molecule along the surface (in our case the centriole surface) is directly related to the rate at which the molecule binds-to/unbinds-from that surface. In such a system, a significant proportion of the molecules that unbind from the surface will inevitably diffuse away rather than rebind to the centriole surface. Calculating the "return probability" of diffusing species in such a system is a complex issue [35,36], however we shall make the simplifying assumption that this "loss" term may be absorbed into the reaction components of the equations (see below).

The two components in such Turing systems are always a slowly diffusing (i.e. rapidly binding and/or slowly unbinding) species termed an Activator ($A$), and a rapidly diffusing (i.e. slowly binding and/or rapidly unbinding) species termed an Inhibitor ($I$), which satisfy

$$\frac{\partial A}{\partial t} = f(A, I) + D_A \frac{\partial^2 A}{\partial x^2} \tag{1}$$

$$\frac{\partial I}{\partial t} = g(A, I) + D_I \frac{\partial^2 I}{\partial x^2} \tag{2}$$

where $t$ is time, $x$ is arc length (i.e. position) along the circular centriole surface, $f$ and $g$ are prescribed functions that describe the chemical reactions that determine how $A$ and $I$ accumulate and decay over time around the centriole surface, and $D_A$ and $D_I$ are the diffusivities of $A$ and $I$ on the centriole surface, respectively. We use this model as our basic framework.

In order for such a general reaction–diffusion system to break symmetry (i.e. to be a Turing system), there are two conditions that the model must satisfy [37]. First, there must exist a steady state (independent of $x$ and $t$) that is stable in the absence of diffusion. Second, this steady state must be unstable in the presence of diffusion. Such a system breaks symmetry when the relatively fast chemical reactions initially drive the system towards the steady state, but this becomes destabilised by diffusion, driving the pattern-forming instability over a longer timescale.

These conditions put constraints on the values of the equation parameters that can support symmetry breaking in Turing systems. Mathematically, these constraints can be expressed for any given system as a set of inequalities that define the possible range of parameter values that will support symmetry breaking. As a consequence of these constraints, symmetry breaking requires that the activator and inhibitor in these systems only interact with each other in one of 2 well-defined regimes (Type I and Type II, Fig 1C). In both regimes, the inhibitor must diffuse more rapidly than the activator in order to drive a process known as short-range activation/long-range inhibition. It is possible to generate (non-Turing) 2 component systems that break symmetry without invoking differential diffusion [38]. In this study by Chau and colleagues, symmetry breaking occurs because the compartments compete for the finite resources

in the system—the total amount of species is fixed, and cytoplasmic depletion is not assumed to be negligible. Such cytoplasmic depletion is almost certainly not occurring during PLK4 symmetry breaking—FRAP experiments, for example, show that PLK4 continuously turns over at centrioles [15,26,39]. An in-depth derivation and analysis of the stability criteria for Turing-type activator–inhibitor systems can be found in [37], and we summarise the key results in Appendix I in S1 Appendix.

## Model 1: Activator to inhibitor conversion based on the model of Takao and colleagues, 2019

The first model we analyse is a Type I system in which a slowly diffusing activator $A$ is converted into a rapidly diffusing inhibitor $I$ via phosphorylation (Fig 2A). As an example, we adapt the biology of the model originally proposed by Takao and colleagues (2019), in which unphosphorylated, kinase inactive, PLK4 is initially recruited to the centriole surface where it self-assembles into slowly turning-over macromolecular condensates. As PLK4 levels in the condensates increase, Takao and colleagues allow PLK4 to auto-phosphorylate itself to create PLK4*, which turns-over more rapidly within the condensates, as observed in vitro [14–16]. This difference in the turn-over rates of the non-phosphorylated and phosphorylated species is the basis for the differential diffusion of the 2 components in the Turing system we formulate here. Takao and colleagues assume that the non-phosphorylated PLK4 self-assembly rate is subject to a sigmoidal attenuation (i.e., as the condensates grow, they become less likely to disassemble) because the central regions of the condensate become progressively more isolated from the cytoplasm as the condensate grows. We also adopt this sigmoidal self-assembly relationship for the production of $A$ (Eq 3). Whereas Takao and colleagues assume that PLK4 ($A$) and PLK4* ($I$) are restricted to binding to a defined number of discrete, spatially organised compartments around the periphery of the centriole, in our model we allow both species to bind freely anywhere on a continuous centriole surface. This model reads:

$$\frac{\partial A}{\partial t} = \frac{aA^2}{1+A} - bA\sqrt{I} - cA + D_A \frac{\partial^2 A}{\partial x^2} \tag{3}$$

$$\frac{\partial I}{\partial t} = bA\sqrt{I} - dI + D_I \frac{\partial^2 I}{\partial x^2}. \tag{4}$$

The first (+ve), middle (-ve), and final (+ve) terms of the equation describe how $A$ (Eq 3) or $I$ (Eq 4) are produced, removed, and diffuse within the system, respectively. The rate parameters $a$, $b$, $c$, and $d$ correspond to the self-assembly rate of the unphosphorylated PLK4 complex (i.e., the rate at which $A$ is produced in the system) ($a$), the phosphorylation rate of PLK4 by phosphorylated PLK4* (i.e., the rate at which $I$ converts $A$ into $I$) ($b$), and the rate at which unphosphorylated PLK4 ($c$) or phosphorylated PLK4* ($d$) are either degraded or lost to the cytoplasm. The precise functional form of these equations, which correspond to the strength of the self-assembly of PLK4 and the strength of *trans*-autophosphorylation of inactive PLK4 by active PLK4*, respectively, are discussed in Appendix II in S1 Appendix.

In general, it is not possible to precisely assign the activator and inhibitor relationships depicted in the schematic in Fig 2A to specific terms in the reaction equations (Eq 3 and Eq 4). This is because, mathematically, a positive feedback for $A$ means simply that $\frac{\partial f}{\partial A} > 0$, and a negative feedback for $I$ means simply that $\frac{\partial g}{\partial I} < 0$; these inequalities may be achieved with complex expressions that extend beyond the usual proportional relationships often assumed. Nevertheless, an approximate description is provided in the figure legend. As discussed above, the conditions for this system to break symmetry take the form of a set of inequalities for the

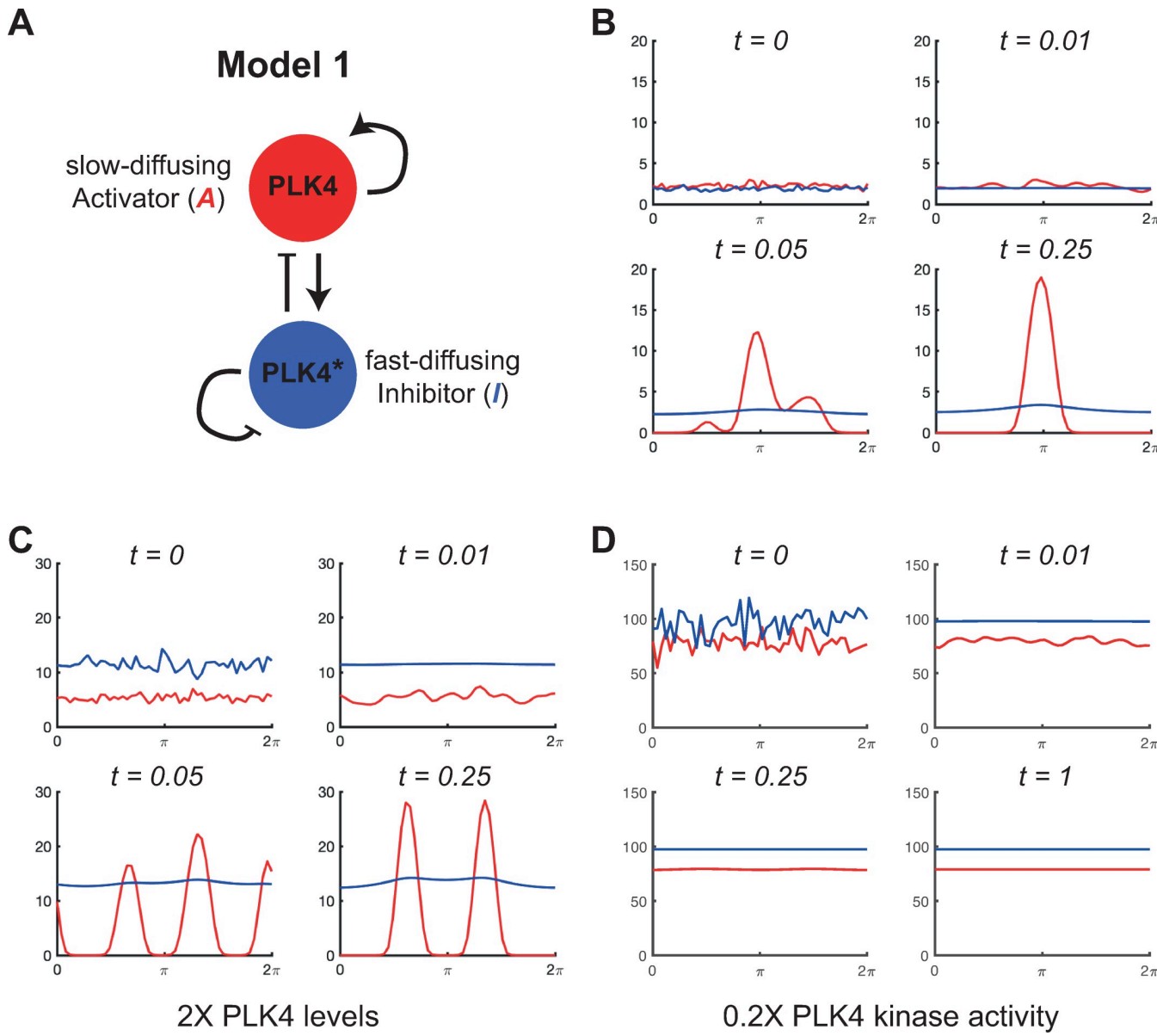

**Fig 2. Computer simulations of mathematical Model 1. (A)** Schematic summarises the reaction regime of Model 1, a Type I Turing system. The biology underlying this model was originally proposed by Takao and colleagues (2019). Although it is not possible to precisely assign the activator and inhibitor relationships depicted in this schematic to specific terms in the reaction equations (Eq 3 and Eq 4, main text) we can approximately see that *A* (non-phosphorylated PLK4, red) promotes the production of more *A* through self-assembly, and the production of *I* (phosphorylated PLK4, denoted PLK4*, blue) by stimulating the self-phosphorylation of *A* as the concentration of *A* increases. *I* inhibits the production of *A* by converting it to *I* via phosphorylation, and *I* inhibits its own accumulation by promoting its own degradation/dissociation. **(B–D)** Graphs show how the levels of the activator species (red lines) and the inhibitor species (blue lines) change over time (arbitrary units) at the centriole surface (modelled as a continuous ring stretched out as a line between 0–2π) in computer simulations. The rate parameters for each simulation are defined in the main text and can lead to symmetry breaking to a single peak (B), or to multiple peaks when cytoplasmic PLK4 levels are increased (C), or to no symmetry breaking at all (with the accumulation of relatively high levels of activator and inhibitor species spread uniformly around the centriole surface) when PLK4 kinase activity is reduced (D). The underlying data for this figure can be found in S1 Data.

parameters *a*, *b*, c, and *d*, which we derive in Appendix II in S1 Appendix. Unless stated otherwise, we choose the parameter values $a = 500$, $b = 250$, $c = 0$, $d = 400$, $D_A = 2$, and $D_I = 5,000$, that satisfy these conditions. These values were chosen in part to reflect the order of magnitude of similar parameter values and ratios used in the previous modelling papers [13,17]. For

example, in the Takao and colleagues manuscript, most reaction parameters have an order of magnitude in the region of 0.01. Assuming units of $\mu$m and $s^{-1}$ and their system timescale of 10 h, this corresponds to a dimensionless value of 360 when modelling the system over a unit timescale; this is consistent with our values chosen for $a$, $b$, and $d$. It is not straightforward to make such a comparison with the Leda and colleagues study, since their parameters range from $10^{-5}$ s to 10 s and there is no direct mapping between their parameters and ours. However, the ratio of the unbinding rates of their phosphorylated-STIL species to unphosphory-lated-STIL species is 0.01. In the Takao and colleagues model, the ratio of diffusion rates is $10^{-4}$. We have chosen an intermediate value of $4 \times 10^{-4}$. Note that this mapping of parameter values between models is necessarily imperfect because the parameters are not describing exactly the same thing in the different models. Importantly, as none of these parameter values are known in any of the models (in our formulation or in their original formulations), the precise values chosen are actually of little significance (as long as, in our modelling, they satisfy the symmetry breaking criteria for a Turing system).

In Fig 2B, we plot the solution of this model (i.e., a computer simulation of how PLK4 ($A$) and PLK4* ($I$) levels vary over time along the centriole surface, from 0 to $2\pi$) subject to the initial conditions $A = A_0(1+W_A(x))$ and $I = I_0(1+W_I(x))$. Here, $A_0$ and $I_0$ are the homogeneous steady-state solutions to (3) and (4), and $W_A$ and $W_I$ are independent random variables with uniform distribution on [0, 1] that we use to generate the initial stochastic noise in the binding of $A$ and $I$ to the centriole surface at $t = 0$. All of the simulations that follow are performed over a unit of dimensionless time ($t = 0$ to $t = 1$), so the timescales of each simulation can be compared. All reaction and diffusion parameters in the system are large compared to unity, so all simulations achieve a steady state within this unit of time. The dimensionless concentration values on the y-axis of the graphs shown in Fig 2B–2D can be compared within these simulations.

We observe that the initial noise in the system is rapidly suppressed for $I$ and begins to be smoothed for $A$ as the system approaches the steady state that would be stable in the absence of diffusion (Fig 2B, $t = 0$ to $t = 0.01$). However, on a slightly longer timescale, this state is destabilised by the presence of diffusion (Fig 2B, $t = 0.05$) as any region in which the levels of $A$ are slightly raised are reinforced due to the self-promotion of $A$. The same effect also increases $I$, but, since $I$ rapidly diffuses away, the local accumulation of $A$ is maintained while the production of $A$ is inhibited around the rest of the centriole surface by the diffusing $I$. The solution therefore rapidly resolves to a nonhomogeneous stable steady state with a single dominant peak (Fig 2B, $t = 0.25$). Note that both $A$ and $I$ continue to dynamically bind/unbind from the centriole in this steady state—in agreement with FRAP experiments showing that PLK4 continuously turns-over at centrioles—but this state is stable and remains unchanged for the remainder of the simulation (i.e., until $t = 1$; not shown). It is also interesting to note that, in this model, it is mostly the activator species (i.e., non-phosphorylated, presumably kinase inactive, PLK4) that accumulates in the single PLK4 peak (Fig 2B), which may seem biologically implausible (see Discussion).

As described in the Introduction, it has been shown experimentally that overexpressing PLK4 leads to the generation of multiple PLK4 foci around the mother centriole, whereas inhibiting PLK4 kinase activity prevents PLK4 symmetry breaking with PLK4 uniformly accumulating to abnormally high levels around the centriole surface. To test if Model 1 could recapitulate these behaviours, we simulated PLK4 overexpression by increasing the PLK4 production rate parameter $a$ by 2-fold and PLK4 kinase inhibition by reducing the phosphorylation rate parameter $b$ by 5-fold. Doubling PLK4 production led to an increase in the number of transient PLK4 peaks that quickly settled to a stable steady state of 2 peaks (Fig 2C). The 2 peaks were evenly spaced around the centriole, which is typical of Turing systems (see

Discussion). In contrast, PLK4 symmetry was no longer broken when PLK4 kinase activity was reduced, and PLK4 accumulated evenly around the centriole to a high steady-state level (Fig 2D). This happens because less $I$ (PLK4*) is produced when PLK4 kinase activity is reduced, so $I$ can no longer suppress the accumulation of $A$ (inactive PLK4) around the centriole. We conclude that, in this parameter regime, Model 1 can capture well 3 key features of PLK4 behaviour at the centriole: (1) breaking symmetry to a single peak under appropriate conditions; (2) breaking symmetry to more than 1 peak when PLK4 is overexpressed; and (3) failing to break symmetry and accumulating high levels of PLK4 when PLK4 kinase activity is inhibited.

## Model 1 robustness analysis

To assess the robustness of Model 1's ability to generate a single PLK4 peak when parameter values are changed, we generated a phase diagram showing the average number of PLK4 peaks generated over 20 simulations (shown in colour code) as we varied the rate of production of PLK4 ($a$) (the equivalent of varying PLK4s cytoplasmic concentration) and PLK4 kinase activity ($b$) (Fig 3A). Parameter values that do not support symmetry breaking are either indicated in dark blue (no PLK4 peaks, PLK4 distributed evenly at high levels around the centriole) or in black (no PLK4 peaks, very little PLK4 present at the centriole). It can be seen that if PLK4 kinase activity drops below a certain level, the system robustly fails to break symmetry and PLK4 accumulates at high levels around the entire centriole surface (dark blue areas, Fig 3A). Thus, Model 1 robustly predicts the symmetric centriolar recruitment of high levels of PLK4 when PLK4 kinase activity is inhibited [15,26,27].

The phase diagram also reveals, however, that this system is not very robust at generating a single PLK4 peak as the amount of PLK4 is varied (i.e., the system cannot reliably produce a single PLK4 peak over a wide range of PLK4 concentrations). There are essentially no values of PLK4 production ($a$) that can reproducibly generate a single PLK4 peak (meaning that 20/20 simulations generated a single peak) that can still reproducibly generate a single PLK4 peak when $a$ is either halved or doubled (Fig 3A, see S1 Fig for a more detailed illustration). Moreover, although increasing the production of PLK4 in this system generates more than 1 peak, it does not readily generate more than 2 to 3 peaks. This seems inconsistent with the biology, as the overexpression of PLK4 can lead to the assembly of up to 6 procentrioles around the mother centriole—see, for example, [40,41]. It is not clear if the original Takao and colleagues model is able to reliably produce a single peak when parameters are halved or doubled as no analysis of the robustness of the model to parameter changes was performed [13]. As we discuss next, however, it seems likely that these are fundamental issues that will affect all models of this Type I system, leading us to propose a plausible solution.

In order for multiple peaks to occur in our model in the overexpressed PLK4 limit, the diffusivity of the activator needs to be sufficiently small so that the multiple PLK4 peaks formed are thin enough to be accommodated around the centriole surface. An examination of the phase diagram comparing the number of PLK4 peaks formed when the diffusion rates of $A$ ($D_A$) and $I$ ($D_I$) are varied in Model 1 (Fig 3B) illustrates this point. We see that, as the diffusivities of $A$ and $I$ decrease, so the system forms increasing numbers of PLK4 peaks (up to 6 in the parameter regime and at the resolution analysed here—but this number can theoretically be even higher in the appropriate parameter regime). This is because, as their diffusivity decreases, $A$ and $I$ spread less efficiently around the centriole, allowing the formation of multiple, thinner, peaks. However, the decreasing diffusivity of $A$ and $I$ necessarily corresponds to a decreasing transfer of "information" between different regions of the centriole. Consequently, as the diffusivity of the PLK4 species decreases, it becomes increasingly difficult for the

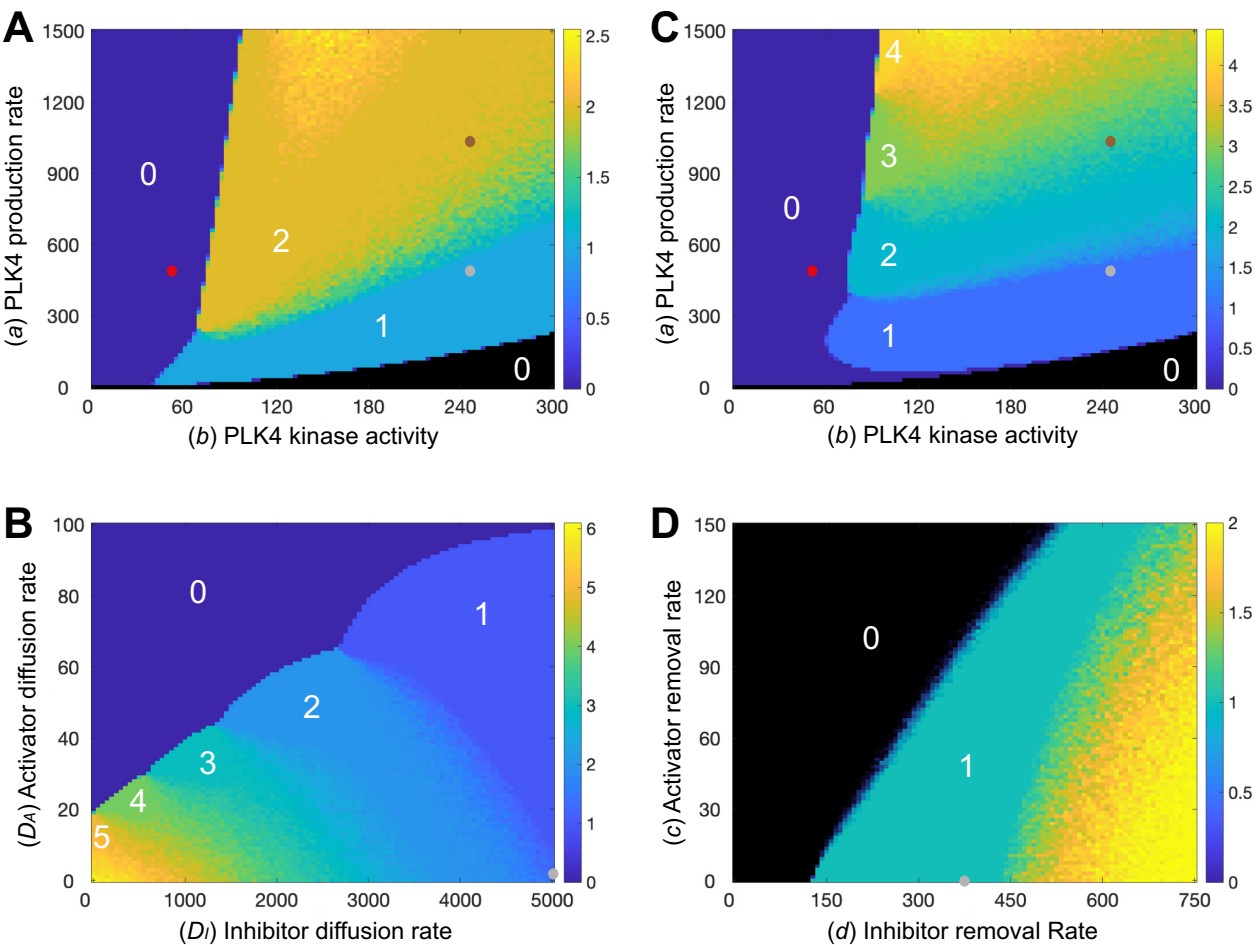

**Fig 3. Analysis of the robustness of Model 1 to changes in parameter values.** Phase diagrams show how the average number of PLK4 peaks generated around the centriole surface in 20 simulations (colour-coded to the scale shown on the right of each diagram) change as different parameters are varied: **(A, C)** The rate of PLK4 production (*a*) and PLK4 kinase activity (*b*) in Model 1 (A), or in a version of Model 1 in which we allow the diffusion rate of the PLK4 species to decrease as the levels of PLK4 in the system increase (C), see text for details. **(B)** The rate of diffusion of the activator ($D_A$) and inhibitor ($D_I$) species. **(D)** The rate at which the activator (*c*) or inhibitor (*d*) species are degraded/lost from the system. The number of peaks formed in certain phase spaces is highlighted (white numbers), and small dots indicate the parameter values used in the simulations shown in Fig 2: normal kinase levels and kinase activity (Fig 2B, grey dots), 2X PLK4 kinase levels (Fig 2C, brown dots), and 0.2X kinase activity (Fig 2D, red dots). Note that brown and red dots are not shown on (B and D) as kinase levels and activity remain constant in the simulations shown in these phase diagrams. The underlying data for this figure can be found in S2 Data.

centriole to robustly form a single peak under normal conditions—since different regions of the centriole are not able to "communicate" with each other efficiently. In other words, single-peak robustness in one limit (low-levels of PLK4) and multiple peaks in another limit (high levels of PLK4) are fundamentally incompatible qualities of the system. Importantly, this issue is not limited to activator–inhibitor/diffusion-based models, but would apply to any spatial model in which information transfers around the centriole surface (as it presumably must do in the real-world physical system).

We realised that the well-characterised ability of PLK4 to dimerise to stimulate its own destruction via *trans*-autophosphorylation [42–45] could potentially solve this problem. This is because any increase in cytoplasmic PLK4 levels will increase the probability of the PLK4 species generated at the centriole surface dimerising and degrading as they diffuse through the bulk cytoplasm. This means that any increase in the cytoplasmic levels of PLK4 will lead to a

reduction in the effective diffusion of $A$ and $I$ around the centriole surface. If we modify Model 1 so that an increase in PLK4 production leads to a decrease in the diffusivity of the PLK4 species (see Appendix IV in S1 Appendix), the system now more robustly forms a single PLK4-peak in the low-PLK4 limit, while simultaneously forming a larger number of peaks in the high-PLK4 limit (Fig 3C; see S1 Fig for a more detailed analysis). Theoretically, the number of peaks that can form is limited only by the lower bound of the diffusivity. In a more general sense, we propose that increasing cytoplasmic PLK4 levels slows down the transfer of information around the centriole due to destructive dimerization; this mechanism permits increasing levels of overduplication in the high-PLK4 limit without affecting system robustness under normal conditions.

Finally, to analyse the effect of varying the dissociation/degradation rates of $A$ ($c$) and $I$ ($d$), we also generated a ($c$, $d$) phase diagram (Fig 3D). We find that the system is able to break symmetry provided that ($c$) is below a certain threshold that depends on ($d$). If ($c$) is above this threshold, then $A$ becomes fully depleted from the system, which in turn removes the source of $I$ (black region, Fig 3D)

## Model 2: Inhibitor to activator conversion based on the model of Leda and colleagues, 2018

The second model we analyse is a Type II system in which a rapidly diffusing inhibitor ($I$) is converted into a slowly diffusing activator ($A$) through phosphorylation. As a biological example of such a model, we adapt the reaction regime proposed by Leda and colleagues (2018). This system comprises just 2 proteins, PLK4 and STIL, but these combine to create 4 components defined by the phosphorylation state of each component, $[PS]$, $[P^*S]$, $[PS^*]$, and $[P^*S^*]$ (* denoting phosphorylation). The authors assume that each component can bind to the centriole surface and may be converted into another, either in the cytoplasm or at the centriole surface, through phosphorylation/dephosphorylation, with the $[P^*S^*]$ species promoting the phosphorylation of all the other species. The centriole complexes in which STIL is phosphorylated are postulated to exchange with the cytoplasm at a slower rate than the complexes in which STIL is not phosphorylated (this will be the basis for the difference in the diffusion rates of the two-component system we formulate below). As in the Takao and colleagues model, Leda and colleagues allow the complexes to bind to discrete centriole compartments, but, importantly, there is no ordering or spatial orientation of these compartments. Instead, compartmental exchange occurs through the shared cytoplasm.

By reinterpreting this cytoplasmic exchange as a diffusive process—with diffusivity dependent on the rate of exchange—we develop a Turing system based on the reactions of the Leda and colleagues model (Model 2; Fig 4A). The 2 relevant components in the system are not PLK4 and STIL, but rather PLK4:STIL complexes that act as either fast-diffusing Inhibitors ($I$) comprising PLK4 bound to non-phosphorylated STIL ($[PS]$ and $[P^*S]$), or slow-diffusing Activators ($A$) comprising PLK4 bound to phosphorylated STIL ($[PS^*]$ and $[P^*S^*]$) (we do this because in the Leda and colleagues model, it is the phosphorylation of STIL that creates the slow-diffusing species, whereas in the Takao and colleagues model, it is the phosphorylation of PLK4 that creates the slow-diffusing species). The derivation of this model is given in detail in Appendix III in S1 Appendix that leads to the system:

$$\frac{\partial A}{\partial t} = bIA^2 - cA + D_A \frac{\partial^2 A}{\partial x^2} \tag{5}$$

$$\frac{\partial I}{\partial t} = a - bIA^2 - dI + D_I \frac{\partial^2 I}{\partial x^2}. \tag{6}$$

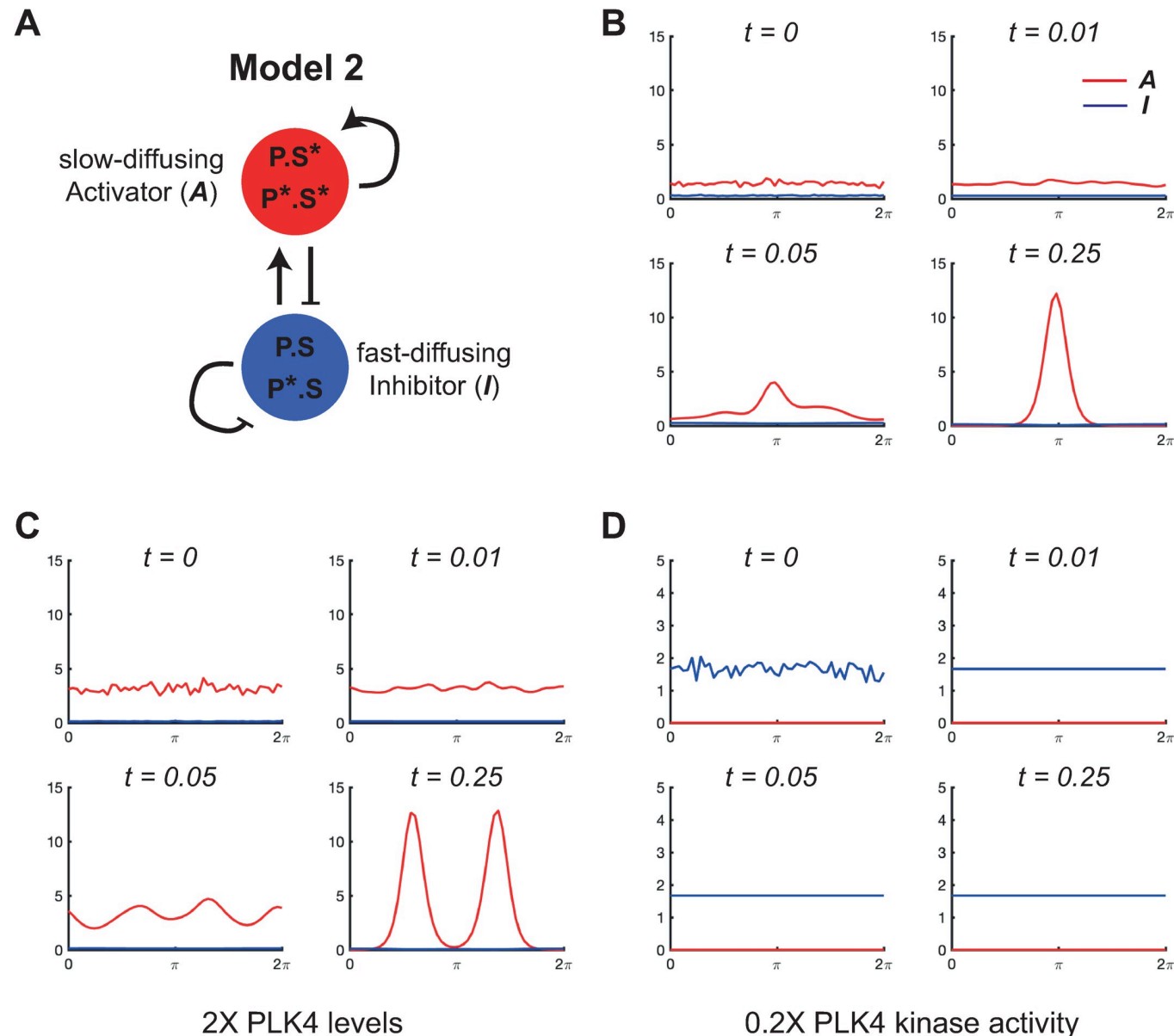

**Fig 4. Computer simulations of mathematical Model 2. (A)** Schematic summarises the reaction regime of Model 2, a Type II Turing system. The biology underlying this model was originally proposed by Leda and colleagues (2018). Although it is not possible to precisely assign the activator and inhibitor relationships depicted in this schematic to specific terms in the reaction equations (Eq 5 and Eq 6, see main text) we can approximately see that *A* (which contains the active P*S* species that phosphorylates all other species) simultaneously generates more *A* and inhibits *I* by converting nearby species that contain non-phosphorylated STIL (*I*) into complexes containing phosphorylated STIL (*A*). *I* promotes the formation of *A* because it acts as the source of *A* in the conversion process, and *I* is self-inhibiting through promoting its own dissociation/degradation (i.e., the rate of dissociation/degradation of *I* is proportional to the amount of *I*—Eq 6). **(B–D)** Graphs show how the levels of the activator species (red lines) and the inhibitor species (blue lines) change over time (arbitrary units) at the centriole surface (modelled as a continuous ring, stretched out as a line between 0–2π) in computer simulations. The rate parameters for each simulation are defined in the main text and can lead to symmetry breaking to a single peak (B), or to multiple peaks when cytoplasmic PLK4 levels are increased (C), or to no symmetry breaking at all—with the near-complete depletion of the activator species from the centriole surface, and the uniform low-level accumulation of the inhibitor species—when PLK4 kinase activity is reduced (D). The underlying data for this figure can be found in S3 Data.

As before, the first (+ve), middle (-ve), and final (+ve) terms of the equation describe how *A* (Eq 5) or *I* (Eq 6) are produced, removed, and diffuse within the system, respectively. Here, *a* is a constant source term for the production of *I* (i.e., the rate at which phosphorylated and

non-phosphorylated PLK4 molecules form complexes with STIL), $b$ is the rate at which $I$ is converted into $A$, (i.e., the rate at which STIL is phosphorylated within these complexes), and $c$ and $d$ are the rates at which the $A$ and $I$ species, respectively, are degraded or lost to the cytoplasm.

As mentioned above, it is not possible to precisely assign the activator and inhibitor relationships depicted in the schematic in Fig 4A to specific terms in the reaction equations (Eq 5 and Eq 6), but an approximate explanation is provided in the figure legend. The conditions for this system to break symmetry are derived in Appendix II in S1 Appendix and, unless stated otherwise, we choose the parameter values $a = 100$, $b = 150$, $c = 60$, $d = 60$, $D_A = 2$, and $D_I = 5{,}000$, that satisfy these conditions. These values were chosen in part to reflect the parameter values and ratios used in the previous modelling papers [13,17] although, as we explain above for Model 1, the precise value of any individual parameter is of little significance.

In Fig 4B, we plot the solution output subject to the initial conditions $A = A_0(1+W_A(x))$ and $I = A_I(1+W_I(x))$, where $A_0$ and $I_0$ are the homogeneous steady-state solutions to (5) and (6) and $W_A$ and $W_I$ are independent random variables with uniform distribution on [0, 1]. As with Model 1, all of the simulations that follow are performed over a unit of dimensionless time and all reaction and diffusion parameters in the system are large compared to unity, so all simulations achieve a steady state within this unit of time. The dimensionless concentration values on the y-axis of the graphs shown in Fig 4B–4D can be compared to each other, but not to the values shown in Fig 2B–2D (for Model 1) as these dimensionless values depend on dimensional reaction rates, which differ between the 2 models.

As with Model 1, the solution approaches a stable nonhomogeneous steady state with a dominant peak after an initial smoothing period. In contrast to Model 1, we observe that, in the region of the activator peak, the inhibitor exhibits a slight dip. This is because the activator promotes the production of the inhibitor in Model 1, but suppresses the production of the inhibitor (by promoting its conversion to the activator) in Model 2. We then simulated PLK4 overexpression in the system by increasing the production rate parameter of the PLK4:STIL complexes ($a$) by 2-fold (Fig 4C) and PLK4 kinase inhibition by reducing the phosphorylation rate parameter ($b$) by 5-fold (Fig 4D). As with Model 1, doubling PLK4:STIL production led to an increase in the number of transient PLK4 peaks that quickly settled to a stable steady state of 2 peaks that were evenly spaced around the centriole. Interestingly, although inhibiting PLK4 kinase activity led to a failure to break symmetry, $A$ was no longer detectable at the centriole and $I$ accumulated only at a low uniform level. Thus, unlike Model 1, Model 2 does not capture well the abnormally high level of uniform accumulation of kinase-inhibited PLK4 species that has been observed experimentally [15,26,27].

## Model 2 robustness analysis

To assess the robustness of Model 2 to changes in parameter values, we first generated a phase-diagram showing the average number of PLK4 peaks generated over 20 simulations as we varied the rate of PLK4:STIL production ($a$) and PLK4 kinase activity ($b$) (Fig 3A). Parameter values that do not support symmetry breaking are indicated in dark blue. In the limit of high PLK4 levels and high kinase activity, the system accumulates high levels of activator uniformly around the centriole (dark blue region, top-right of Fig 3B). This is analogous to all compartments being occupied in a discrete model and would likely result in multiple daughter centrioles being produced. By contrast, at low levels of kinase activity, there is a low-level uniform distribution of the inhibitor and no accumulation of the activator (dark blue region, bottom left of Fig 3B). Therefore, Model 2 robustly fails to recapitulate the abnormal high-level centriolar accumulation of PLK4 observed when PLK4 kinase activity is inhibited [15,26,27]. Thus, in this parameter regime at least, Model 2 cannot be correct.

Interestingly, in this parameter regime, Model 2 also suffers from the same 2 problems we encountered with Model 1: (1) There is no value of *a* that can robustly generate a single peak of PLK4 that can still do this when (*a*) is halved or doubled (S1 Fig); and (2) the system struggles to generate >3–4 PLK4 peaks when PLK4 is overexpressed. As before, decreasing the diffusivity of *A* and *I* allows the system to generate more, thinner, peaks (Fig 5B), and linking the increase in PLK4 production to a decrease in diffusivity (as we did for Model 1—see Appendix IV in S1 Appendix) allows the system to generate more (>6) centrioles when PLK4 is overexpressed, although it has a less pronounced effect on the robustness of the system to produce a single PLK4 peak (Figs 5C and S1). We note that the original Leda and colleagues model [17] avoids these problems because it supposes no spatial relationship between the individual compartments and instead assumes that communication between compartments is instantaneous. However, this requires that diffusion is sufficiently fast that concentration gradients are negligible between centriolar compartments, but not so fast that the relevant species are diluted in the cytoplasm. It seems implausible that both of these effects could be achieved with a single diffusion rate in a real-world physical system where the centrioles exist in the context of a much larger volume of cytoplasm.

The (*c*, *d*) phase diagram (Fig 5D) shows that, provided (*c*) is above a certain threshold, there is a large region of the parameter space in which the system breaks symmetry to a single peak. For values of (*c*) below this threshold, the species accumulate uniformly around the centriole due to the low degradation rate. In contrast, for sufficiently large values of (*c*) and (*d*), both species are fully depleted from the system.

## Unifying the models of Takao and colleagues and Leda and colleagues

At a first glance, the original models of Takao and colleagues and Leda and colleagues appear to be very different: they use different mathematical methods to describe different chemical reactions, with symmetry breaking in the former being driven by PLK4 phosphorylation and in the latter by the phosphorylation of STIL in complexes with PLK4. In deriving Model 2, we grouped together the species $A = [PS^*]+[P^*S^*]$ and $I = [PS]+[P^*S]$ based on the unbinding rates of the 4 species specified in the Leda and colleagues paper—where complexes containing phosphorylated STIL ([$PS^*$] and (*c*) [$P^*S^*$]) unbind slowly and those containing non-phosphorylated STIL ([$PS$] and [$P^*S^*$]) unbind rapidly. It is simple, however, to modify these model parameters so that the unbinding rate now depends on the phosphorylation state of PLK4 in these complexes (as is the case in the Takao and colleagues model), rather than on the phosphorylation state of STIL—i.e., we allow [$P^*S$] and [$P^*S^*$] to now unbind rapidly and [$PS$] and [$PS^*$] to now unbind slowly. Thus, in this reinterpretation, we are essentially applying the biological justification of the Takao and colleagues model to the Leda and colleagues model.

In this scenario, if we set $A = [PS]+[PS^*]$ and $I = [P^*S]+[P^*S^*]$ then, following the same procedure as before (see Appendix III in S1 Appendix), we arrive at a new model,

$$\frac{\partial A}{\partial t} = a - bAI^\alpha - cA + D_A \frac{\partial^2 A}{\partial x^2} \qquad (7)$$

$$\frac{\partial I}{\partial t} = bAI^\alpha - dI + D_I \frac{\partial^2 I}{\partial x^2}. \qquad (8)$$

We observe that by setting $\alpha = 1/2$ and substituting the sigmoidal self-assembly source term $aA^2/(1+A)$ in place of the constant source term, *a*, we obtain exactly Model 1. In other words, all of the dynamics of the Takao and colleagues model is contained within the Leda and colleagues model, but with additional complexity and a different choice of rate parameters.

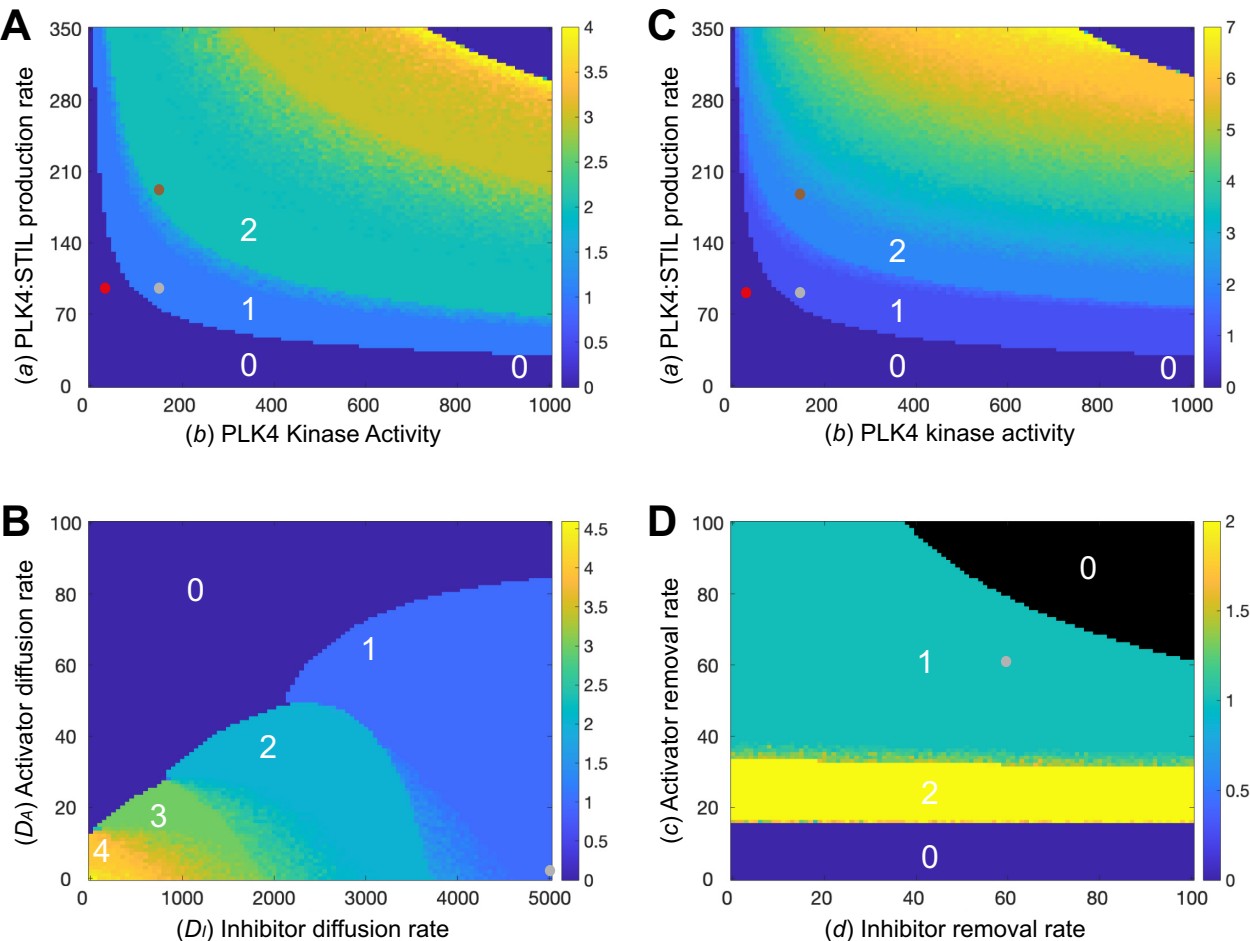

**Fig 5. Analysis of the robustness of Model 2 to changes in parameter values.** Phase diagrams show how the average number of PLK4 peaks generated around the centriole surface in 20 simulations (colour-coded to the scale shown on the right of each diagram) change as different parameters are varied: **(A, C)** The rate of PLK4 production (*a*) and PLK4 kinase activity (*b*) in Model 2 (A), or in a version of Model 2 in which we allow the diffusion rate of the PLK4 species to decrease as the levels of PLK4 in the system increase (C), see text for details. **(B)** The rate of diffusion of the activator ($D_A$) and inhibitor ($D_I$) species. **(D)** The rate at which the activator (*c*) or inhibitor (*d*) species are degraded/lost from the system. The number of peaks formed in certain phase spaces is highlighted (white numbers), and small dots indicate the parameter values used in the simulations shown in Fig 4: normal kinase levels and kinase activity (Fig 4B, grey dots), 2X PLK4 kinase levels (Fig 4C, brown dots), and 0.2X kinase activity (Fig 4D, red dots). Note that brown and red dots are not shown on (B and D) as kinase levels and activity remain constant in the simulations shown in these phase diagrams. The underlying data for this figure can be found in S4 Data.

## Modelling symmetry breaking on a compartmentalised centriole surface

In our modelling so far, PLK4 symmetry breaking occurs on a continuous centriole surface; it does not require that the centriole surface be divided into discrete compartments that effectively compete with each other to become the dominant site (as is assumed in the previous models). Importantly, however, our modelling still applies if we instead divide the centriole surface into an arbitrary number of discrete compartments—with the various PLK4 species interacting only within an individual compartment, but diffusing laterally between the spatially separated compartments (see Appendix V in S1 Appendix). In Fig 6, we show the system outputs of Model 1 (Fig 6A) and Model 2 (Fig 6B) when we run simulations with a centriole surface comprising 9 discrete compartments that interact with the various PLK4 species. In both instances, the systems robustly break their initial symmetry to produce a single dominant

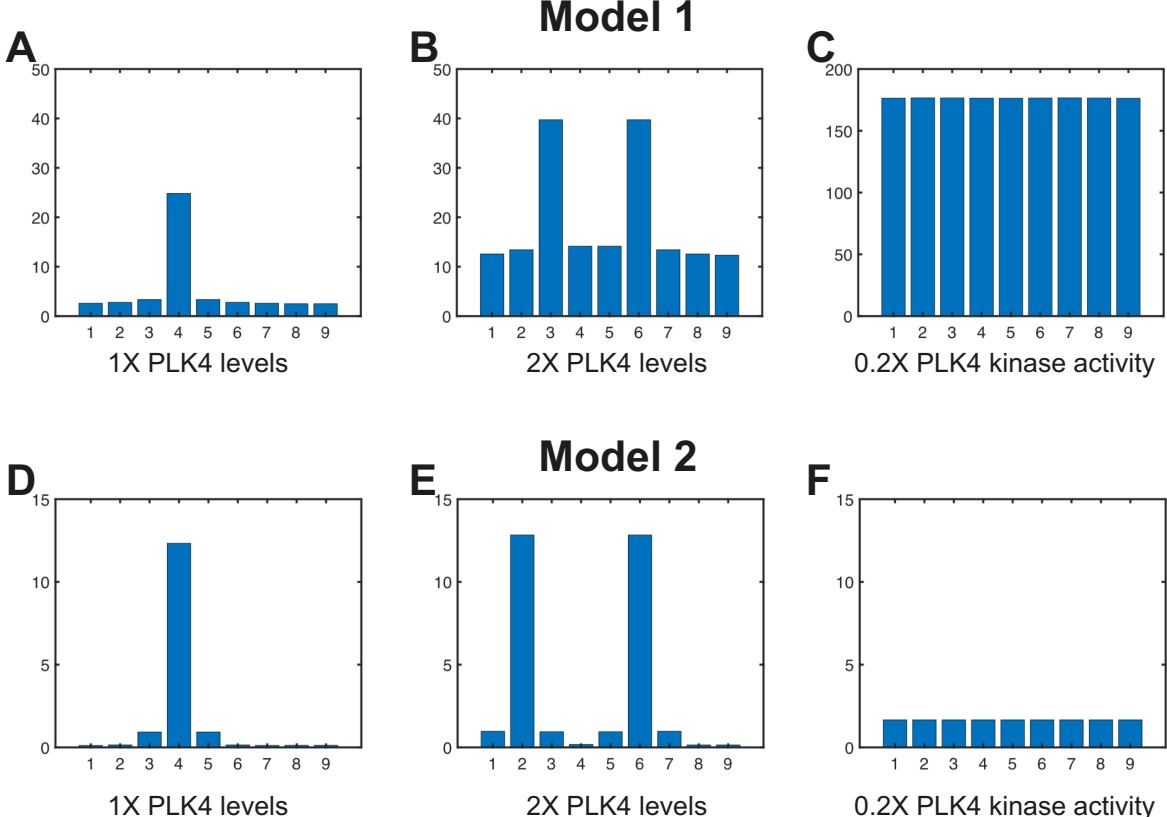

**Fig 6. Analysis of PLK4 symmetry breaking on a centriole surface comprising 9 equally spaced discrete compartments.** Bar charts show the output of computer simulations of Model 1 **(A–C)** and Model 2 **(D–F)** that have been adjusted so that the centriole is no longer 1 continuous surface but rather 9 discrete compartments. In this formulation of our Turing system models, the reactions between all species occur within the individual compartments, but species can diffuse between the 9 compartments that are arranged spatially around the centriole surface. Each compartment is depicted here as an individual bar distributed equally around the circumference of the centriole, but displayed here along a straight line. The height of each bar represents the amount of activator and inhibitor present in that compartment at the end of the simulation when the system has reached a steady state. The underlying data for this figure can be found in S5 Data.

compartment that accumulates the relevant species. Moreover, these discrete models also capture the predicted behaviour of the system when PLK4 is overexpressed (Fig 6C and 6D) or when PLK4 kinase activity is reduced (Fig 6E and 6F). Importantly, these results will hold for any number of compartments (as long as there are at least 2 compartments in the system).

## Discussion

PLK4 is the master regulator of centriole biogenesis, so understanding how it becomes concentrated at a single site on the side of the mother centriole is crucial to understanding how centriole number is so strictly controlled [5,6,46,47]. Here, we mathematically modelled PLK4 symmetry breaking as a simple two-component Turing system. All such models can be classified as either Type I or Type II systems depending on their reaction dynamics [29,37] (Fig 1C), and we presented a Type I model based on the reaction scheme originally proposed by Takao and colleagues (2019) and a Type II model based on the reaction scheme originally proposed by Leda and colleagues (2019). Although Takao and colleagues noted that their system had some similarities to a Turing model, their model is not a reaction–diffusion model, and it does not generate the single PLK4 peak using the property of short-range activation/long-range inhibition that is central to all Turing systems.

By reformulating the previous models in this way, we can now compare them. Surprisingly, although they used different mathematical methods and proposed different reaction regimes, we show that the biology underlying both models can be described by the same Turing system dynamics. Thus, although it was not appreciated at the time, both previous PLK4 symmetry breaking models have arrived at the same solution—with phosphorylated and non-phosphorylated species of PLK4 (either on its own or in a complex with STIL) effectively acting as either a slow-diffusing activator (*A*) or a fast-diffusing inhibitor (*I*) species—with the differential centriole binding/unbinding properties of these species effectively allowing them to diffuse around the centriole at different rates. It seems likely, therefore, that this mechanism underlies PLK4 symmetry breaking.

It is possible to break symmetry in non-Turing systems. For example, Chau and colleagues [38] investigate a related system in which 2 species diffuse laterally around a membrane and exchange with a well-mixed cytosol. They examined all possible network configurations of linear positive/negative feedback to determine the topologies that permit symmetry breaking. By allowing non-negligible exchange with a well-mixed cytosol, they observe that symmetry breaking occurs in more topologies than would be observed for a strict Turing model. However, an advantage of the Turing framework adopted in our manuscript is that it can be mathematically proven that all such systems that break symmetry must satisfy either the Type I or Type II criteria (Fig 1C), and that there are well-defined mathematical constraints on the parameter values required to support symmetry breaking in these systems. Even so, we suspect that it will be challenging to precisely identify the molecular species that comprise *A* and *I* and to precisely define the chemical relationship between them. This is because the interactions of PLK4 with itself [48–53], with its centriole receptors [39,48,54–58], and with other crucial centriole duplication proteins (such as STIL/Ana2) [19,59–63] are complex. PLK4 therefore probably binds to the centriole surface through a web of interactions involving several species. For this reason, we have not attempted to make predictions about the molecular identity of *A* and *I* or their precise reaction-regimes. Nevertheless, by probing these interactions in a general setting, it may be possible to establish several properties that *A* and *I* must satisfy without explicitly knowing their composition, which in turn may help in identifying them. Indeed, our modelling reveals some interesting features of the PLK4 symmetry-breaking system that we discuss below.

First, our studies highlight some potential problems with the biology underlying both previous models of PLK4 symmetry breaking. In Model 1 (based on Takao and colleagues), phosphorylated species of PLK4 act as the fast-diffusing inhibitor, so it is the non-phosphorylated (and so presumably kinase inactive) PLK4 species that will usually accumulate to the highest levels within the PLK4 peak (Fig 2B). A priori, this seems biologically implausible, as PLK4 kinase activity is thought to be essential for centriole assembly [64] because it phosphorylates STIL [65–67]. Model 2 (based on Leda and colleagues) does not suffer from this problem (as the slow-diffusing phosphorylated species will usually accumulate within the PLK4 peak), but it predicts that inhibiting PLK4 kinase activity will lead to the loss of the slowly turning over phosphorylated species from the centriole (Fig 4D). This is clearly inconsistent with data showing that inhibiting PLK4 kinase activity leads to the high-level accumulation of slowly turning-over forms of PLK4 [15,26].

It is currently unclear how to resolve these issues, but we think it worth considering the possibility that it is the inactive form of PLK4 that mostly accumulates in the PLK4 peak (as predicted by Model 1). Perhaps this does not matter, as in Type I systems some inhibitor (i.e., phosphorylated kinase) still accumulates in this peak (Fig 2B), and this may be sufficient to phosphorylate STIL to stimulate daughter centriole assembly. Alternatively, perhaps PLK4 kinase activity is required primarily for PLK4 symmetry breaking, but not canonical daughter

centriole assembly. This may seem heretical, but there is some evidence to support this possibility. For example, in some systems PLK4 kinase activity appears to only be required at the end of mitosis/early G1—when PLK4 symmetry is presumably being broken, but before daughter centrioles have physically started to assemble [68]. Moreover, when co-overexpressed in early *Drosophila* embryos, Sas-6 and STIL/Ana2 assemble into large particles (SAPs) that recruit many other centriole proteins [69,70]. SAP assembly does not require PLK4, yet it appears to be stimulated by the phosphorylation of STIL/Ana2 [70], suggesting that another kinase could phosphorylate STIL/Ana2 to promote daughter centriole assembly.

Second, a feature of these Turing models is that multiple peaks will become evenly spaced within the system if given enough time to do so, as peaks are most stable when they are as far apart from each other as possible. (It should be noted, however, that the process that drives this even spacing may not be significant enough to overcome the noise inherent to the system, and will be influenced by any inhomogeneities in the substrate.) Unfortunately, the spacing of multiple PLK4 peaks in cells overexpressing PLK4 has not been quantified, so one is left to interpret published images of multiple peaks, some of which might support equal spacing while others appear not to [40,41]. This analysis is complicated, however, as PLK4's main centriolar receptor, CEP152/Asl, can form incomplete or undulating rings around the mother centriole in some cell types [26,27]. Thus, it will be important to measure PLK4-peak spacing in relation to the underlying CEP152/Asl ring.

Third, it was difficult to find parameters in our models that supported the robust selection of a single PLK4 site under normal conditions while still supporting the generation of >2–3 PLK4 peaks when PLK4 was overexpressed (S1 Fig). This was surprising, as PLK4 normally very robustly generates a single daughter centriole, while its overexpression can generate 6 to 7 daughter centrioles [40,41]. This problem occurs in our models because the diffusion rate of the inhibitor needs to be fast enough to inhibit the formation of multiple peaks around the centriole under normal conditions, but slow enough to allow the formation of >2–3 peaks when PLK4 is overexpressed. This problem is therefore likely to apply to any model in which information has to be communicated around the centriole surface. We found that this problem can be at least partially solved if we allow PLK4's diffusion rate to decrease as its concentration in the cytoplasm increases (Figs 3C and 5C). This seems biologically justifiable, as any PLK4 molecule diffusing around the centriole is more likely to dimerise with another PLK4 molecule —and so be targeted for *trans*-autophosphorylation-dependent degradation [42,44,45,71]—if the cytoplasmic concentration of PLK4 is higher. Even with this modification, however, the ability of our models to generate a single PLK4 site was surprisingly sensitive to variations in PLK4 concentration. This may reflect the real physical situation, as even a modest increase in PLK4 concentration (of probably no more than approximately 2-fold) can trigger centriole overduplication [25,72]. This may explain why cytoplasmic PLK4 levels appear to be kept so unusually low [25,73,74].

A final potential advantage of our modelling approach is that it does not rely on PLK4 being recruited to centrioles by receptors that are organised into discrete compartments that compete with each other for PLK4 binding, as was the case in both previous models. While there is some data supporting the idea that CEP152/Asl may be organised into discrete compartments, the number and organisation of these compartments is unclear [13,21–24,75]. Moreover, it has been proposed that CEP152, together with its binding partner CEP63, can assemble into a continuous ring around the centriole [76,77]. Intriguingly, the most recent expansion microscopy and super-resolution studies suggest that CEP152 may provide a "flexible" binding surface around the mother centriole that is not uniform, but is not highly compartmentalised either [26,27]. Thus, reality may lie somewhere between the discrete compartment and uniformly homogeneous limits, which is compatible with our modelling approach.

## Supporting information

**S1 Appendix. Description of Mathematical Methods.**
(DOCX)

**S1 Fig. Analysing system robustness if the diffusion rate of the activator and inhibitor species is allowed to decrease as the concentration of PLK4 increases. (A)** The phase diagram shown here is the same as that shown in Fig 3A comparing Model 1's robustness to changes in the rate of PLK4 production and kinase activity. The magenta boxes each highlight a particular concentration of PLK4 (thick horizontal line) and the vertical height of the boxes show the values when that concentration is halved (bottom of boxes) or doubled (top of boxes). It can be seen that in this original version of Model 1, there is essentially no concentration of PLK4 that can robustly generate a single PLK4 peak (light blue area) that can still robustly generate a single PLK4 peak when its concentration is either halved or doubled. Also, when these concentrations of PLK4 are increased further, they do not readily generate more than 2 to 3 PLK4 peaks. **(B)** The phase diagram shown here is the same as that shown in Fig 3C, where we adapted Model 1 to allow the diffusivity of the PLK4 species to decrease as the concentration of PLK4 increases. In this adapted model, it can be seen that the concentrations of PLK4 highlighted with the magenta horizontal lines and associated boxes can still robustly generate a single PLK4 peak when that concentration of PLK4 is either halved or doubled and would also more readily form multiple peaks of PLK4 when PLK4 is overexpressed to even higher levels. **(C, D)** Phase diagrams show the same analysis of Model 2 as described in (A and B) above for Model 1. The underlying data for this figure can be found in S6 Data.
(EPS)

**S1 Data. Data for the plots presented in Fig 2.** The 3 sheets in the.xlsx file corresponds to the 3 subfigures (Fig 2B–2D). Each sheet contains 4 pairs of columns, with each pair of columns corresponding to a single subplot of the figure, given in chronological order. Within each pair, the first column contains the data for the activator, $A$, and the second column contains the data for the inhibitor, $I$, at that time point.
(XLSX)

**S2 Data. Data for the plots presented in Fig 3.** The.xlsx file contains 4 sheets corresponding to the 4 plots displayed in Fig 3 (Fig 3A–3D). Each sheet contains a $101 \times 101$ matrix, with columns corresponding to $x$-axis values and rows corresponding to $y$-axis values in the appropriate parameter space (shown in Fig 3A–3D). The numerical value at each point in the matrix reflects the number of peaks observed at that point in parameter space. A NaN entry reflects depletion of the species in addition to no symmetry breaking.
(XLSX)

**S3 Data. Data for the plots presented in Fig 4.** The 3 sheets in the.xlsx file corresponds to the 3 subfigures (Fig 4B–4D). Each sheet contains 4 pairs of columns, with each pair of columns corresponding to a single subplot of the figure, given in chronological order. Within each pair, the first column contains the data for the activator, $A$, and the second column contains the data for the inhibitor, $I$, at that time point.
(XLSX)

**S4 Data. Data for the plots presented in Fig 5.** The.xlsx file contains 4 sheets corresponding to the 4 plots displayed in Fig 5 (Fig 5A–5D). Each sheet contains a $101 \times 101$ matrix, with columns corresponding to $x$-axis values and rows corresponding to $y$-axis values in the appropriate parameter space (shown in Fig 5A–5D). The numerical value at each point in the matrix reflects the number of peaks observed at that point in parameter space. A NaN entry reflects

depletion of the species in addition to no symmetry breaking.
(XLSX)

**S5 Data. Data for the plots presented in Fig 6.** The file contains 6 sheets corresponding to the 6 plots displayed in Fig 6 (Fig 6A–6F). Each sheet contains a single column vector of length 9 which reflects the values in the bar chart of the corresponding subfigure.
(XLSX)

**S6 Data. Data for the plots presented in S1 Fig.** The.xlsx file contains 4 sheets corresponding to the 4 plots displayed in S1 Fig (S1A–S1D Fig). Each sheet contains a $101 \times 101$ matrix, with columns corresponding to $x$-axis values and rows corresponding to $y$-axis values in the appropriate parameter space (shown in S1A–S1D Fig). The numerical value at each point in the matrix reflects the number of peaks observed at that point in parameter space. A NaN entry reflects depletion of the species in addition to no symmetry breaking.
(XLSX)

## Author Contributions

**Conceptualization:** Zachary M. Wilmott, Alain Goriely, Jordan W. Raff.

**Formal analysis:** Zachary M. Wilmott, Alain Goriely, Jordan W. Raff.

**Funding acquisition:** Jordan W. Raff.

**Investigation:** Zachary M. Wilmott.

**Methodology:** Zachary M. Wilmott.

**Project administration:** Jordan W. Raff.

**Writing – original draft:** Zachary M. Wilmott, Jordan W. Raff.

**Writing – review & editing:** Zachary M. Wilmott, Alain Goriely, Jordan W. Raff.

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
