## [Editor Report · Decision Letter 0]

24 Jul 2023

Dear Dr Raff,

Thank you for submitting your revised manuscript from Review Commons, entitled "A simple Turing reaction-diffusion model can explain how mother centrioles break symmetry to generate a single daughter" for consideration as a Research Article by PLOS Biology. 

Your manuscript has now been evaluated by the PLOS Biology editorial staff, as well as by an academic editor with relevant expertise, and I am writing to let you know that we would like to pursue your manuscript further and send it back out for re-review by the original reviewers at Review Commons.

However, before we can send your manuscript back to the reviewers, we need you to complete your submission by providing the metadata that is required for full assessment. To this end, please login to Editorial Manager where you will find the paper in the 'Submissions Needing Revisions' folder on your homepage. Please click 'Revise Submission' from the Action Links and complete all additional questions in the submission questionnaire.

Once your full submission is complete, your paper will undergo a series of checks in preparation for peer review. After your manuscript has passed the checks it will be sent out for review. To provide the metadata for your submission, please Login to Editorial Manager (https://www.editorialmanager.com/pbiology) within two working days, i.e. by Jul 26 2023 11:59PM.

Kind regards,

Richard

Richard Hodge, PhD

rhodge@plos.org

PLOS

---

## [Decision Letter · Decision Letter 1]

25 Aug 2023

Dear Dr Raff,

Thank you for your patience while we considered your revised manuscript "A simple Turing reaction-diffusion model can explain how mother centrioles break symmetry to generate a single daughter" for consideration as a Research Article at PLOS Biology. Your revised study has now been evaluated by the PLOS Biology editors, the Academic Editor and the original reviewers at Review Commons. 

The reviews are pasted below my signature. As you will see, the reviewers note that the manuscript is improved and Reviewer #1 specifically notes that the revision has convinced him/her about the significance of the model in relation to the previous Takao and Leda models. On the other hand, Reviewer #2 raises concerns with the overall strength of advance and asks for several reporting details and additional discussions. This includes explicitly outlining the previous problems with the Takao and Leda models and clarifying the rationale for the parameter choices. In addition, Reviewer #1 asks that the previous findings in Chau et al (Cell, 2012) are appropriately contextualized and discussed and Reviewer #3 notes that a specific statement in the manuscript should be amended. 

In light of the reviews, we are pleased to offer you the opportunity to address the remaining points from the reviewers in a revision that we anticipate should not take you very long. We will then assess your revised manuscript and your response to the reviewers' comments with our Academic Editor aiming to avoid further rounds of peer-review, although might need to consult with the reviewers, depending on the nature of the revisions.

In addition, I would be grateful if you could make sure to address the following data and other policy-related requests that I have provided below (A-E):

(A) We would like to suggest the following modification to the title: 

““A simple Turing reaction-diffusion model explains how PLK4 breaks symmetry during centriole duplication and assembly”

(B) You may be aware of the PLOS Data Policy, which requires that all data be made available without restriction: http://journals.plos.org/plosbiology/s/data-availability. For more information, please also see this editorial: http://dx.doi.org/10.1371/journal.pbio.1001797

-Supplementary files (e.g., excel). Please ensure that all data files are uploaded as 'Supporting Information' and are invariably referred to (in the manuscript, figure legends, and the Description field when uploading your files) using the following format verbatim: S1 Data, S2 Data, etc. Multiple panels of a single or even several figures can be included as multiple sheets in one excel file that is saved using exactly the following convention: S1_Data.xlsx (using an underscore).

-Deposition in a publicly available repository. Please also provide the accession code or a reviewer link so that we may view your data before publication. 

Figure 2B-D, 3A-D, 4B-D, 5A-D, S1A-D

(C) Please also ensure that each of the relevant figure legends in your manuscript include information on *WHERE THE UNDERLYING DATA CAN BE FOUND*, and ensure your supplemental data file/s has a legend.

(D) Per journal policy, as the code that you have generated is important to support the conclusions of your manuscript, we require that you make it available without restrictions upon publication. Please ensure that the code is sufficiently well documented and reusable, and that your Data Statement in the Editorial Manager submission system accurately describes where your code can be found.

(E) Please ensure that your Data Statement in the submission system accurately describes where your data can be found.

**IMPORTANT - SUBMITTING YOUR REVISION**

*Resubmission Checklist*

*Published Peer Review*

*PLOS Data Policy*

*Blot and Gel Data Policy*

Sincerely,

Richard

Richard Hodge, PhD

rhodge@plos.org

REVIEWS:

Reviewer #1: The authors convinced me with their description of the novelty of the model and its significance with respect to previous models. I also read the authors' response to other reviewers and was also convinced that the description of a symmetry break in the Turing reaction diffusion mechanism may be of interest beyond the centriole duplication community.

I have considered the authors' argument about the novelty of their findings compared to the work of Chau et al. (Cell 2012). However, I still believe that the final reaction scheme they propose in Figure 4 was already predicted in that publication to break symmetry and polarise cells with two interacting components. The key to this mechanism was not cytoplasmic depletion but local concentration. I appreciate that in Chau et al the components do not permanently detach and relocalise, which PLK4 does, and so the components concentrate to complete cytoplasmic depletion, which PLK4 does not, but the interaction scheme is still the same. The dynamics are different, but the mechanism is the same. So for me, the conceptual innovation of this model is limited and the work of Chau and Lim could be discussed more than just in the introduction. However, it is valuable to apply it to the example of PLK4 dynamics.

Reviewer #2: Review of the revised manuscript by Dr. Jordan W. Raff for PLOS Biology:

"A simple Turing reaction-diffusion model can explain how mother centrioles break symmetry to generate a single daughter."

We thank the authors for their rebuttal document and manuscript revisions, and appreciate the improvements that have been made on the analysis of the models. As a result, the reformulations of the two existing models can be better understood, thereby facilitating comparisons. However, we remain unconvinced that a continuum reformulation of these models reflects Plk4 symmetry breaking more accurately, or that such reformulation increases our understanding of the process in another way. Therefore, we remain of the opinion that the work bears insufficient novelty and significance to justify publication as a research manuscript. Instead, after the points listed below have been addressed, a revised manuscript would be well suited as a commentary piece or a review on reaction-diffusion systems explaining Plk4 symmetry breaking at the centriole. 

We think that the following major comments have not been fully addressed in the revision:

Comments of Reviewer #1

In the rebuttal document, the authors raise numerous problems with the previous studies of Takao et al. and Leda et al., mentioning for instance "both have significant issues", "some very odd behaviours emerge", "the bizarre manner in which Plk4 overexpression drives the formation of multiple PLK4 peaks" and "The authors do not comment on, analyse, or explain these strange phenomena." These issues are not mentioned in the revised manuscript itself, let alone analyzed or explained from the perspective of the models in their original formulation. Such concerns should be spelled out explicitly in the revised manuscript if readers are to understand what prompted the authors to initiate their reformulation.

In addition, one of the statements about the model by Leda et al. appears to be wrong. The authors mention that the original Leda et al. model predicts that inhibiting Plk4 kinase activity leads to Plk4 depletion around the centriole. This might be the case in the authors' two-component reformulation of the Leda et al. model, but not in the original model, in which Plk4-STIL (PS) uniformly accumulates in all compartments when Plk4 kinase activity is inhibited (i.e. setting the rates at which PS transitions into P*S and PS* to 0). 

Comments of Reviewer #2 (points refer to the numbering in the original review)

1. We thank the authors for expanding their analysis and finding a model adaptation enabling the generation of multiple peaks in the Plk4 overexpression limit. In order to do so, the authors have postulated a mechanism in which Plk4 diffusivity around the torus decreases with cytoplasmic Plk4 concentration. As a result, the width of a Plk4 peak becomes smaller as the number of peaks increases. This expanded analysis prompts the following questions: 

(a) How much thinner do Plk4 peaks become? In Model 1, the maximum number of peaks that can be achieved robustly seems to be 5 (and in some cases 6, Figure 3B). 

(b) Is there a theoretical limit to the number of peaks formed? 

(c) If so, how does the maximum number of peaks relate to the 9 peaks in which Plk4 is reportedly coalescing upon inhibition of Plk4 kinase activity (mentioned in the Discussion on p. 27)? If there is not theoretical limit to the maximum number of peaks, why should there be only 9 peaks experimentally?

(d) More generally, could the 9 peaks observed upon Plk4 kinase inhibition invalidate the use of formulating Plk4 symmetry breaking as a Turing-based two-component reaction-diffusion system on a continuous line? 

(e) Can multiple peaks be generated in discretized versions of the model? And does the discretization allow for multiple peaks upon Plk4 overexpression without the requirement for the peaks to become thinner, or does the set compartment size prevent forming multiple peaks? This is an important point to clarify.

(f) Furthermore, the authors state that the inability to form multiple peaks is a fundamental issue that would be present in any spatial model (bottom of p. 14), but state also that "it is not clear if these problems also occur in the original formulations of the Takao et al. model as no analysis of the robustness of the model to parameter changes was performed" (top of p. 14). Both Leda et al. and Takao et al. models form multiple peaks in overexpression conditions without peaks becoming thinner as their number increases. What convinces the authors that their mechanism is a better representation of the formation of multiple peaks upon Plk4 overexpression?

2. We thank the authors for adopting our suggestion to discuss the strong prediction their reformulation of the models makes on the spacing of multiple Plk4 peaks. However, we do not agree with their statement that this prediction cannot be compared to that of the Leda et al. model. In the rebuttal document, the authors correctly note that "Leda et al. assume no spatial relationship between PLK4-binding compartments." From this, the authors infer that the model does not make predictions about the spacing of multiple Plk4 peaks, stating that "peak-spacing cannot be assessed." Instead, we interpret that a strong prediction from the lack of spatial relationship in Leda et al. is that multiple peaks have no dependence on each other, and thus should be randomly positioned relative to each other. 

Points 4-6. We asked the authors to clarify their parameter choices. From the rebuttal document and the revised manuscript, it remains unclear how the specific parameter values were chosen. In the revised manuscript, the authors mention that "values were chosen in part to reflect values and ratios in the previous modelling papers when adjusted to the same timescale", without specifying how. For the reader's interest, the authors should make it clear how values of the original models are mapped to specific parameter values in their reformulations, and specify which values in the reformulations originate from which values in the original models. This could be integrated for instance in the Appendix pertaining to the reformulated models. As stated already during our initial review, we think that the authors should discuss if any other set of values that satisfies the derived mathematical requirements would result in the same qualitative outcomes with similar robustness. Alternatively, potential empirical reasons why these values are preferred should be mentioned explicitly. Furthermore, it is unclear why the specific parameter values chosen are different for the two models. This is especially puzzling as it is then shown that the two models can be written in the same terms. Whether the qualitative outcome of the models is analogous when using the same parameters for both models are used should be mentioned. If this is not the case, an explanation should be provided.

In response to our comments in the 'significance' section, in which we had noted that Takao et al. already mentioned that a Turing model can explain symmetry breaking at the centriole, the authors revised the manuscript by mentioning that "Takao et al. described their system as being analogous to a 'Turing model', but it is not a reaction-diffusion model" (p. 23). Here, the authors refer to the lateral-inhibition model, which is analogous to reaction-diffusion model, but indeed not one per se. However, as the authors themselves mention in the rebuttal document but not in the revised manuscript, Takao et al. also performed an analysis of the reaction-diffusion version of their model. Additionally, both the lateral-inhibition and the reaction-diffusion versions of the original Takao et al. model consisted of two components. Therefore, the fact that a two-component Turing reaction-diffusion model can explain how mother centrioles break PLK4 symmetry to generate a single daughter is already evident from Takao et al.'s work.

Reviewer #3: The authors have made several improvements to the revised version of the manuscript and have addressed most of my concerns. The formulation of PLK4 symmetry breaking into a Turing reaction system still yields inconsistencies with the experimental evidence. Nevertheless, the authors are open about these limitations and thoroughly discuss them. The revised manuscript will interest centriole enthusiasts and those modeling biological systems. 

* The authors state: 'An alternative possibility is that PLK4 kinase activity is only required for PLK4 symmetry breaking and not daughter centriole assembly.' In my opinion, the cited evidence does not strongly support this possibility. The high overexpression of centriole proteins may drive some centriole reactions, but at physiological concentrations, PLK4 is always required to make new centrioles. Even de novo centriole assembly requires PLK4 kinase activity, and presumably, there is no symmetry breaking required for de novo assembly. Moreover, there are direct PLK4 phosphorylation sites that have been identified on STIL. Preventing phosphorylation of these sites blocks centriole assembly.

---

## [Editor Report · Decision Letter 2]

29 Sep 2023

Dear Dr Raff,

Thank you for your patience while we considered your revised manuscript "A simple Turing reaction-diffusion model can explain how mother centrioles break symmetry to generate a single daughter" for publication as a Research Article at PLOS Biology. This revised version of your manuscript has been evaluated by the PLOS Biology editors and the Academic Editor.

Based on our Academic Editor's assessment of your revision, I am pleased to say that we are likely to accept this manuscript for publication. Whilst we appreciate the points you make in the rebuttal regarding the additional discussions for the Chau et al and Takao et al studies, we ask that you please include the folllowing three minor clarifications in the manuscript text to provide further contextualization for the model for a broad readership. 

1. Briefly revisit the relevance of the Chau et al. model when discussing the novelty/merits of the approach in the Discussion.

2. Clarify how, if the parameters used in the Leda et al. and Takao et al. models cannot be extracted, they can be used to inform the parameters (or their magnitude of their differences) used in the current model.

3. For the benefit of the reader, briefly mention in the Discussion the evidence in references 68 and 69 that is consistent with the heretical idea.

*IMPORTANT*

In addition, I would be grateful if you could make sure to address the following data and other policy-related requests that I have provided below (A-E):

(A) We would like to suggest the following modification to the title:

““A simple Turing reaction-diffusion model explains how PLK4 breaks symmetry during centriole duplication and assembly”

(B) You may be aware of the PLOS Data Policy, which requires that all data be made available without restriction: http://journals.plos.org/plosbiology/s/data-availability. For more information, please also see this editorial: http://dx.doi.org/10.1371/journal.pbio.1001797

-Supplementary files (e.g., excel). Please ensure that all data files are uploaded as 'Supporting Information' and are invariably referred to (in the manuscript, figure legends, and the Description field when uploading your files) using the following format verbatim: S1 Data, S2 Data, etc. Multiple panels of a single or even several figures can be included as multiple sheets in one excel file that is saved using exactly the following convention: S1_Data.xlsx (using an underscore).

-Deposition in a publicly available repository. Please also provide the accession code or a reviewer link so that we may view your data before publication.

Figure 2B-D, 3A-D, 4B-D, 5A-D, S1A-D

(C) Please also ensure that each of the relevant figure legends in your manuscript include information on *WHERE THE UNDERLYING DATA CAN BE FOUND*, and ensure your supplemental data file/s has a legend.

(D) Per journal policy, as the code that you have generated is important to support the conclusions of your manuscript, we require that you make it available without restrictions upon publication. Please ensure that the code is sufficiently well documented and reusable, and that your Data Statement in the Editorial Manager submission system accurately describes where your code can be found.

(E) Please ensure that your Data Statement in the submission system accurately describes where your data can be found.

We expect to receive your revised manuscript within two weeks. 

*Published Peer Review History*

*Press*

Sincerely,

Richard

Richard Hodge, PhD

rhodge@plos.org

PLOS

---

## [Editor Report · Decision Letter 3]

18 Oct 2023

Dear Dr Raff,

Thank you for the submission of your revised Research Article "A simple Turing reaction-diffusion model explains how PLK4 breaks symmetry during centriole duplication and assembly" for publication in PLOS Biology. On behalf of my colleagues and the Academic Editor, Simon Bullock, I am pleased to say that we can accept your manuscript for publication, provided you address any remaining formatting and reporting issues. These will be detailed in an email you should receive within 2-3 business days from our colleagues in the journal operations team; no action is required from you until then. Please note that we will not be able to formally accept your manuscript and schedule it for publication until you have completed any requested changes.

In addition, please note that I took the liberty of changing the title of the manuscript in the manuscript file to the one that we had suggested during the previous round of review. Thank you for letting us know that you were happy with the new title and had changed it, but it seemed that the title was unchanged in our editorial software.

PRESS

Best wishes, 

Richard

Richard Hodge, PhD

rhodge@plos.org

PLOS
